



# Observed wavenumber-frequency spectrum of global, normal mode function decomposed, fields: a possible evidence for nonlinear effects on the wave dynamics

Andre S. W. Teruya[1], Breno Raphaldini[2], Victor C. Mayta[3], Carlos F. M. Raupp[1], and Pedro L. da Silva Dias[1]

[1]Instituto de Astronomia, Geofísica e Ciências Atmosféricas, Universidade de São Paulo, Sao Paulo, Brazil.
[2]Department of Mathematical Sciences, Durham University, Durham, UK.
[3]Department of Climate and Space Science and Engineering, University of Michigan, Ann Arbor, Michigan, and Department of Atmospheric and Oceanic Sciences, University of Wisconsin-Madison , Madison, United States.

**Correspondence:** Andre Teruya (andre.teruya@usp.br)

**Abstract.** The study of tropical tropospheric disturbances has led to important challenges from both observational and theoretical points of view. In particular, the observed wavenumber-frequency spectrum of tropical oscillations, also known as Wheeler-Kiladis diagram, has helped bridging the gap between observations and the linear theory of equatorial waves. Here we have obtained a similar wavenumber-frequency spectrum for each equatorial wave type by performing a normal mode function (NMF) decomposition of global Era-Interim reanalysis data, with the NMF basis being given by the eigensolutions of the primitive equations in spherical coordinates, linearized around a resting background state. In this methodology, the global multi-level horizontal velocity and geopotential height fields are projected onto the normal mode functions characterized by a vertical mode, a zonal wavenumber, a meridional quantum index and a mode type, namely Rossby, Kelvin, mixed Rossby-gravity and westward and eastward propagating inertio-gravity modes. The horizontal velocity and geopotential height fields associated with each mode type are then reconstructed on the physical space, and the corresponding wavenumber-frequency spectrum is calculated for the 200 hPa zonal wind. The results reveal some expected structures, such as the dominant global-scale Rossby and Kelvin waves constituting the intraseasonal frequency associated with the Madden-Julian Oscillation. On the other hand, some unexpected features such as westward propagating Kelvin waves and eastward propagating westward inertio-gravity waves are also revealed by our observed 200 hPa zonal wind spectrum. These intriguing behaviours represent a large departure from the linear equatorial wave theory and can be a result of strong nonlinearities in the wave dynamics.

## 1   Introduction

The study of tropical tropospheric oscillations is one of incredible complexity due to the interaction of multiple scales, strong nonlinearities and multiple competing physical phenomena (Khouider et al., 2012), including the Madden-Julian Oscillation





(MJO) (Zhang, 2005), tropical convection (Kiladis and Weickmann, 1992), convectively coupled waves (Kiladis et al., 2009), interactions with the extra-tropics (Ferranti et al., 1990), interactions with the stratosphere (Raphaldini et al., 2020d), tropical cyclones (Emanuel, 2003), tropical rainfall (Gadgil, 2003; Mayta et al., 2020), among others, each of these topics with crucial socio-economic and environmental impacts.

From the theoretical framework, some of the first successful attempts of tropical meteorology originated from the modelling
of waves on the equatorial beta-plane, going back to the works of Matsuno and others in the 1960s (Matsuno, 1966; Lindzen, 1967; Yanai and Maruyama, 1966). The waves arising in this type of system, either shallow water or primitive equations (Majda, 2003; Gill, 1982), on the equatorial beta plane, have their mid-latitude counterparts such as the Rossby and inertio-gravity waves, but also some modes that are unique of the equatorial region arise such as Kelvin and mixed Rossby-Gravity modes. The latter modes were recently shown to have deep theoretical roots due to topological invariants, behaving as interfacial waves
between different media (Delplace et al., 2017).

An important step towards the connection between the equatorial wave theory and observations was initially achieved in Takayabu (1994a), Pires et al. (1997) and Wheeler and Kiladis (1999). These studies obtained the wavenumber vs. time frequency spectrum of tropical oscillations from either the dynamical field variables or the outgoing long-wave radiation (OLR), revealing a striking correspondence between the observed time frequencies and the theoretical predictions from the linear equa-
torial wave theory. These and subsequent studies (e.g., Wheeler et al., 2000) have been regarded as observational evidence for the existence of convectively coupled equatorial waves. In the simplest theories, the convectively coupled waves are oscillations that resemble the free linear modes, but with slower frequencies due to the coupling of the waves with moist convection; these theories are reviewed in Kiladis et al. (2009). Besides the observed oscillations that exhibit a good correspondence with the linear (free or convectively coupled) equatorial waves, Kiladis et al. (2009) also gave insight into the structure of the Madden-
Julian Oscillation, which does not correspond to any of the linear equatorial wave modes (Zhang, 2005). The correspondence between the time-frequency spectrum of observed OLR or velocity field oscillations with the linear eigenfrequency of equatorial waves suggest that one may isolate the field variables associated with a single wave type by filtering out the corresponding band in the wavenumber-frequency spectrum. This type of decomposition has been carried out in a number of studies (e.g., Wheeler et al., 2000). In this context, departures from the linearized theory may have several origins such as a non-resting
basic state (Yang et al., 2003; Dias and Kiladis, 2014), more complex mechanisms of moisture coupling (Khouider and Majda, 2006, 2007) as well as nonlinear effects (Khouider et al., 2012) and the coupling with oceanic processes (Ramirez et al., 2017).

An alternative, and mathematically more rigorous, approach to decompose a given set of velocity and pressure fields into their corresponding eigenmodes is to project them onto the orthogonal basis set given by the equatorial wave eigenfunctions. This procedure was carried out, for instance, in Gehne and Kleeman (2012) using the equatorial beta-plane shallow water
model. Despite the simplicity of their model equations, they show some important results such as the role of long barotropic Rossby waves in generating spectral peaks on synoptic scales. A normal mode decomposition approach combined with space-time spectral analysis was also put forth by Castanheira and Marques (2015), who studied the coherence between the OLR signal and the normal-mode decomposed fields.



In the present study, we use a similar procedure to those documented in Gehne and Kleeman (2012) and Castanheira and

Marques (2015), in which the contribution of each mode type to the field variables is obtained from the projection onto the corresponding mode eigenvector rather than the spectral peak bands in the wavenumber-frequency domain. In this context, as in Castanheira and Marques (2015), the basis function set associated with the modal decomposition is given by the eigensolutions of the spherical geometry primitive equations linearized around a resting background state. In this way, the corresponding normal mode functions (NMF) consist of the eigenfunctions of a rigid lid boundary condition Sturm-Liouville problem as

vertical structure functions and the eigenfunctions of the Laplace's tidal equations, the so-called Hough verctor harmonics (Longuet-Higgins, 1968; Kasahara, 1976, 1977), as the horizontal structure functions. We use ERA-Interim reanalysis data from the European Centre for Medium-Range Weather Forecasts (Dee et al., 2011) to decompose the velocity and geopotential height fields onto the NMF basis. The projection procedure was first proposed by Kasahara and Puri (1981); here we use a particular implementation of this procedure provided by the open source MODES software described in Žagar et al. (2015).

This procedure has been successful in describing different atmospheric phenomena such as the MJO (Žagar and Franzke, 2015) and the QBO (Raphaldini et al., 2020c, d).

After the normal mode decomposition of the horizontal velocity and geopotential height fields, we obtained the wavenumber-frequency spectrum of the reconstructed 200 hPa zonal velocity field associated with each mode type, namely Rossby (R), Kelvin (K), mixed Rossby-gravity (MRG) and eastward/westward propagating inertio-gravity modes (EIG/WIG), using the

methodology proposed by Kiladis et al. (2009). The results reveal some surprising departures from the expected linear wave theory. For instance, waves propagating in the opposite direction to what is expected from the linear theory (e.g., WIG waves propagating eastward), as well as significant departures from the linear eigenfrequencies. We argue that these discrepancies can be attributed to the role of strong nonlinearities in the wave dynamics.

## 2 Normal mode function and equatorial wave theory

### 2.1 Normal mode function

As previously mentioned, the basis functions utilized here for the projection of the 3-dimensional global field data set are the normal modes of the compressible primitive equations in the spherical coordinate frame, linearized around a resting background state. Following the formulation of Kasahara and Puri (1981), these equations can be written using the so-called $\sigma$-coordinate system as follows:

$$\frac{\partial u}{\partial t} - 2\Omega \sin(\phi)v = -\frac{1}{a\cos(\phi)}\frac{\partial P}{\partial \lambda}, \tag{1}$$

$$\frac{\partial v}{\partial t} + 2\Omega \sin(\phi)u = -\frac{1}{a}\frac{\partial P}{\partial \phi}, \tag{2}$$

$$\frac{\partial}{\partial t}\left[\frac{\partial}{\partial \sigma}\left(\frac{\sigma}{R\Gamma_0}\frac{\partial P}{\partial \sigma}\right)\right] - \nabla \cdot \mathbf{V} = 0, \tag{3}$$





In the equations above, $\mathbf{V} = (u, v)$ refers to the horizontal wind field, with $u$ and $v$ indicating its zonal and meridional components, respectively, and $\sigma = p/p_s$ is the vertical coordinate, where $p$ and $p_s$ are the pressure and surface pressure fields, respectively; $(\lambda, \phi)$ refers to the regular longitude-latitude coordinate system, $\Omega$ is the Earth rotation rate, $R$ the dry air gas constant, $a$ the average Earth's radius, $\Gamma_0$ the static stability parameter of the stably stratified background atmosphere and

$$P = gz + RT_0(\sigma) \ln p_s,$$

with $T_0$ corresponding to the background temperature, z the geopotential height and $g$ the gravity acceleration. Assuming the rigid lid boundary conditions $\frac{d\sigma}{dt} = 0$ at $\sigma = 0$ and at $\sigma = 1$, Kasahara and Puri (1981) showed that the eigensolutions of system
(1)-(3) above can be expressed as:

$$\begin{bmatrix} u(\lambda, \phi, \sigma, t) \\ v(\lambda, \phi, \sigma, t) \\ P(\lambda, \phi, \sigma, t) \end{bmatrix} = \mathbf{X}_{m,k,n}^{(\alpha)}(\phi) \exp[ik\lambda - i\omega_{m,k,n}^{(\alpha)}t] G_m(\sigma) \tag{4}$$

In (4), $k$ is the zonal wavenumber, $m$ and $n$ are the indices that characterize the vertical and meridional structures of the eigenmodes, respectively, and the index $\alpha$ distinguishes the wave type, as will be discussed later. In this context, the vertical structure functions $G_m(\sigma)$ satisfy the following Sturm-Liouville problem

$$\frac{d}{d\sigma}\left(\frac{\sigma g}{R\Gamma_0}\frac{dG_m}{d\sigma}\right) + \frac{1}{H_m}G_m = 0 \tag{5}$$

$$\frac{dG_m}{d\sigma} = 0 \text{ at } \sigma = 0 \tag{6}$$

$$\frac{dG_m}{d\sigma} + \frac{\Gamma_0}{T_0}G_m = 0 \text{ at } \sigma = 1 \tag{7}$$

where the separation constant $H_m$ is labeled as equivalent height (Taylor, 1936) and is determined from the eigenvalues of the boundary-value problem above. To find these eigenvalues and their corresponding eigenfunctions, we have used the numerical
procedure proposed in Kasahara and Puri (1981), which consists of solving the matrix eigenvalue problem obtained from the finite-difference representation of equations (5)-(7). The solutions corresponding to the first 40 eigenvalues are displayed in Fig. 1. One notices that the first 5 modes exhibit a barotropic structure in the troposphere, that is, they do not change sign throughout this region. In addition, the higher the vertical mode index m, the more oscillatory structure the eigenfunction exhibits. In particular, the last 10 modes are highly oscillatory in the troposphere but have only weak oscillations in the
stratosphere.

On the other hand, the meridional structure vector function $\mathbf{X}_{m,k,n}^{(\alpha)} = [u_{m,k,n}^{(\alpha)}(\phi), iv_{m,k,n}^{(\alpha)}(\phi), gh_{m,k,n}^{(\alpha)}(\phi)]^T$, also known as Hough vector functions (Hough, 1898; Kasahara, 1976, 1977), and the eigenfrequencies $\omega_{m,k,n}^{(\alpha)}$ satisfy the following eigen-





value problem

$$-\omega_{m,k,n}^{(\alpha)} u_{m,k,n}^{(\alpha)} - 2\Omega\sin(\phi) v_{m,k,n}^{(\alpha)} + \frac{gkh_{m,k,n}^{(\alpha)}}{a\cos\phi} = 0, \tag{8}$$

$$\omega_{m,k,n}^{(\alpha)} v_{m,k,n}^{(\alpha)} + 2\Omega\sin(\phi) u_{m,k,n}^{(\alpha)} + \frac{g}{a}\frac{dh_{m,k,n}^{(\alpha)}}{d\phi} = 0, \tag{9}$$

$$\omega_{m,k,n} h_{m,k,n}^{(\alpha)} + \frac{H_m}{a\cos\phi}\left[ ku_{m,k,n}^{(\alpha)} + \frac{d(v_{m,k,n}^{(\alpha)}\cos\phi)}{d\phi} \right] = 0, \tag{10}$$

together with the assumption that $h_{m,k,n}^{(\alpha)} = 0$ at the poles. The eigenvalue problem presented above, also known as Laplace's tidal equations, can only be solved numerically, unless asymptotic approximations are made as in Longuet-Higgins (1968). In the open-source MODES software utilized here, the eigenfrequencies and the corresponding Hough vector eigenfunctions $\mathbf{X}_{m,k,n}^{(\alpha)}$ are obtained from the numerical procedure described in Kasahara (1976) and Swarztrauber and Kasahara (1985). According to Kasahara (1976), the eigensolutions of (8)-(10) can be divided into the symmetric and antisymmetric eigenmodes. For the symmetric modes, $u_{m,k,n}^{(\alpha)}$ and $h_{m,k,n}^{(\alpha)}$ exhibit an even symmetry about the equator and $v_{m,k,n}^{(\alpha)}$ an odd symmetry, whereas the antisymmetric modes are characterized by $u_{m,k,n}^{(\alpha)}$ and $h_{m,k,n}^{(\alpha)}$ exhibiting an odd symmetry and $v_{m,k,n}^{(\alpha)}$ an even one. As will be shown later, the symmetric (antisymmetric) modes are labeled by an odd (even) meridional quantum index n. Longuet-Higgins (1968) classified the mode types of (8)-(10) as the first kind oscillations, which correspond to the high-frequency westward/eastward propagating inertio-gravity waves, and the second kind oscillations or rotational modes representing the so-called Rossby-Haurwitz waves. The Kelvin wave corresponds to the first eastward propagating symmetric mode of the first kind oscillations, while the mixed Rossby-gravity waves refer to the first antisymmetric rotational mode (Kasahara, 1976).

## 2.2 Equatorial wave theory

Although we have utilized the normal mode functions on the sphere for the eigenmode decomposition of the observed large-scale atmospheric fields, since the Hough vector functions can only be obtained numerically, it is useful to make an analogy with the equatorial wave theory in order to show approximate analytical expressions for the meridional structure functions $\mathbf{X}_{m,k,n}^{(\alpha)}$. In fact, for small values of the equivalent height $H_m$, the Hough vector functions can be approximated by the eigensolutions of the equatorial $\beta$-plane version of (8)-(10) (Gill, 1982). These eigensolutions of the equatorial $\beta$-plane shallow-water equations can be expressed in terms of the orthogonal basis of parabolic cylinder functions (Matsuno, 1966; Majda, 2003; Gill, 1982). The meridional structure of the non-dispersive[1] Kelvin wave is written as:

$$\mathbf{X}_{m,k}^{(K)} = \begin{bmatrix} e^{-\xi^2/2} \\ 0 \\ e^{-\xi^2/2} \end{bmatrix}. \tag{11}$$

---

[1]The equatorial Kelvin wave dispersion relation is given by $\omega_{m,k}^{(K)} = k\sqrt{gH_m}$.





where

$$\xi = \frac{y}{\left(\frac{\sqrt{gH_m}}{\beta}\right)^{\frac{1}{2}}},$$

with y representing the meridional displacement from the equator and $\beta = 2\Omega/a$ the equatorial value of the Rossby parameter. The mixed Rossby-gravity (MRG) wave and the first anti-symmetric eastward propagating inertio-gravity wave (EIG) have meridional structure functions of the form:

$$\mathbf{X}_{m,k,0}^{(\alpha)} = \begin{bmatrix} \dfrac{\xi e^{-\xi^2/2}}{\omega_{m,k,0}^{(\alpha)} + k\sqrt{gH_m}} \\[2em] i e^{-\xi^2/2} \\[2em] \dfrac{\xi e^{-\xi^2/2}}{\omega_{m,k,0}^{(\alpha)} + k\sqrt{gH_m}} \end{bmatrix}, \tag{12}$$

with the eigenfrequencies $\omega_{m,k,0}^{(\alpha)}$ satisfying

$$\omega_{m,k,0}^{(\alpha)} = k\sqrt{gH_m}\left[\frac{1}{2} \pm \frac{1}{2}\left(1 + \frac{4\beta}{k^2\sqrt{gH_m}}\right)^{\frac{1}{2}}\right] \tag{13}$$

where the negative (positive) sign in (13) refers to $\alpha = MRG$ ($\alpha = EIG$).

For the other EIG, as well as for the Rossby (R) and westward propagating inertio-gravity (WIG) waves, the meridional structure functions are given by

$$u_{m,k,n}^{(\alpha)} = \frac{1}{2}(\omega_{m,k,n}^{(\alpha)} - kc_m)D_{n+1}(\xi) + n(\omega_{m,k,n}^{(\alpha)} + kc_m)D_{n-1}(\xi) \tag{14}$$

$$v_{m,k,n}^{(\alpha)} = i((\omega_{m,k,n}^{(\alpha)})^2 - k^2 c_m^2)D_n(\xi) \tag{15}$$

$$h_{m,k,n}^{(\alpha)} = \frac{1}{2}(\omega_{m,k,n}^{(\alpha)} - kc_m)D_{n+1}(\xi) - n(\omega_{m,k,n}^{(\alpha)} + kc_m)D_{n-1}(\xi) \tag{16}$$

where $D_n(\xi) = H_n(\xi)e^{-\xi^2/2}$ is the Hermite function, with $H_n(\xi)$ representing the $n-$th degree Hermite polynomial[2], $c_m = \sqrt{gH_m}$ and the eigenfrequencies $\omega_{m,k,n}^{(\alpha)}$ satisfy the dispersion relation

$$\frac{(\omega_{m,k,n}^{(\alpha)})^2}{gH_m} - k^2 - \frac{k\beta}{\omega_{m,k,n}^{(\alpha)}} = (2n+1)\frac{\beta}{\sqrt{gH_m}}.$$

## 3  Data and methods

We have used the Era-Interim (ERAI) reanalysis data from the European centre of Medium Weather Forecast (Dee et al., 2011) for the 1980-2019 period. The data set consists of horizontal velocity and geopotential height fields, and its spatial resolution is of $2.5^o \times 2.5^o$.

---

[2]$H_n(\xi)$ are defined such that $\int_{-\infty}^{\infty} H_m(y)H_n(y)e^{-y^2}dy = \delta_{m,n}\sqrt{\pi}2^n n!$





**Figure 1.** Vertical structure functions corresponding to the first 40 vertical modes of the normal mode decomposition.

### 3.1 Normal Mode Decomposition

From the orthogonality and completeness of the normal mode functions described in Section 2.1, the observed atmospheric fields can be expanded in a normal mode function series as follows:

$$
\begin{bmatrix} u(\lambda,\phi,\sigma,t) \\ v(\lambda,\phi,\sigma,t) \\ P(\lambda,\phi,\sigma,t) \end{bmatrix} = \sum_{m=0}^{+\infty} \sum_{k=-\infty}^{+\infty} \sum_{n=0}^{+\infty} \sum_{\alpha} C_{m,k,n}^{(\alpha)}(t) \mathbf{X}_{m,k,n}^{(\alpha)}(\phi) e^{ik\lambda} G_m(\sigma) \tag{17}
$$

In this way, given the observations of the horizontal winds, pressure and geopotential height fields, the spectral coefficients $C_{m,k,n}^{(\alpha)}(t)$ are obtained by the projection of the observed fields onto a particular eigenmode characterized by a vertical mode index m, a zonal wavenumber k, a meridional quantum index n and a mode type labeled by $\alpha$:

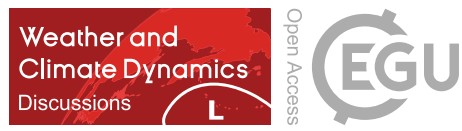

$$C_{m,k,n}^{(\alpha)}(t) = \int\limits_0^1 \int\limits_0^{2\pi} \int\limits_{-\pi/2}^{\pi/2} \left[ u(\lambda,\phi,\sigma,t)u_{m,k,n}^{(\alpha)}(\phi) + v(\lambda,\phi,\sigma,t)v_{m,k,n}^{(\alpha)}(\phi) + P(\lambda,\phi,\sigma,t)gh_{m,k,n}^{(\alpha)}(\phi) \right] e^{-ik\lambda} G_m(\sigma)a^2 \cos(\phi)d\phi d\lambda d\sigma$$

(18)

The decomposition illustrated above is performed using the open source MODES software (Žagar et al., 2015). Details on the numerical procedures are provided in Žagar et al. (2015) and Kasahara and Puri (1981). The coefficients $C_{m,k,n}^{(\alpha)}(t)$ obtained from (18) can be used to compute filtered reconstructions of the atmospheric fields; for instance, the reconstruction
of the dynamical fields associated with a single mode type:

$$\begin{bmatrix} u^{(\alpha)}(\lambda,\phi,\sigma,t) \\ v^{(\alpha)}(\lambda,\phi,\sigma,t) \\ P^{(\alpha)}(\lambda,\phi,\sigma,t) \end{bmatrix} = \sum_{m=0}^{+\infty} \sum_{k=-\infty}^{+\infty} \sum_{n=0}^{+\infty} C_{m,k,n}^{(\alpha)}(t)\mathbf{X}_{m,k,n}^{(\alpha)}(\phi)e^{ik\lambda}G_m(\sigma)$$

(19)

### 3.2 Wheeler-Kiladis diagram

The Wheeler-Kiladis diagram corresponds to the frequency-wavenumber spectrum of tropical disturbances. This technique, initially documented in Takayabu (1994a) and then in Wheeler and Kiladis (1999), constructs the diagram based on OLR
signals. However, since normal mode decomposition comprises only velocity and geopotential height fields, we chose the zonal component of the velocity field (u) at the 200hPa level to construct the Wheeler-Kiladis diagram. The implementation was made using NCAR command language available at https://www.ncl.ucar.edu/Document/Functions/Diagnostics/wkSpaceTime.shtml, NCL (2019). The procedure consists in performing the Fourier Transform regarding both zonal and time dependencies of the corresponding field averaged within the equatorial belt. This procedure results in a raw spectrum, which is usually difficult
to interpret. Thus, Wheeler and Kiladis (1999) proposed to remove the background red-noise spectrum from the calculated spectrum in order to emphasize local maximum signals throughout the $k - \omega$ space. In this context, Wheeler and Kiladis (1999) defined the background spectrum as a smoothed version of the raw spectrum obtained by applying a 1-2-1 filter to the raw spectrum in the $k - \omega$ space. The Wheeler-Kiladis diagram results from the division of the the raw spectrum at each point $(k, \omega)$ by the smoothed background spectrum.

## 4 Results

For illustrative purposes, Figure 2 shows the 200hPa horizontal wind and geopotential fields at 12UTC of January 1st, 2001. Figure 2(a) displays the original fields, while the remaining panels illustrate their filtered versions for specific wave types according to equation (19). The flow pattern displayed in Fig. 2(a) is clearly dominated by the subtropical jet streams having planetary-scale waves embedded. As this flow pattern is of rotational character, with relative small divergence, it is essentially
due to the contribution of the rotational modes, as can be clearly demonstrated by the comparison of Figs. 2(a) and (b). Figure



**Figure 2.** Snapshot of the horizontal velocity and geopotential fields at 200 hPa (a) and their decomposition onto the normal mode function basis for rotational modes (b), the baroclinic component of the rotational modes (c), eastward inertio gravity (EIG) modes (d), westward inertio-gravity modes (e), Kelvin wave (f) and the mixed Rossby-Gravity mode (g).





2(c) is the same as Fig. 2(b) but with the vertical modes having a barotropic structure in the troposphere ($m = 1, 2, 3, 4, 5$) being excluded. The flow pattern illustrated in Fig. 2(c) is still dominated by the subtropical jets, but with equally important contributions in some regions of the tropics, most notably in the Pacific sector. Figures 2(d) and (e) present the contribution of eastward and westward inertio-gravity waves, respectively, to the horizontal wind and geopotential fields displayed in Panel

(a). As a consequence of the dominance of the rotational modes to the flow pattern illustrated in Fig. 2(a), the dynamical fields due to the activity of the inertio-gravity modes is of much weaker magnitude, as can be observed in Figs. 2(d) and (e). It is also clear in Figs. 2(d) and (e) the divergent character of the flow associated with the high-frequency waves. The Kelvin wave contribution is presented in figure 2(f). As expected, the Kelvin component is characterized by a symmetric about the equator zonal flow in phase with the geopotential field, with both exhibiting a strong trapping near the equator. The mixed Rossby-

gravity mode contribution displayed in Fig. 2(g) is also strongly confined near the equator, but exhibiting a strong symmetric about the equator meridional flow and an antisymmetric about the equator geopotential field.

### 4.1 The Wheeler-Kiladis diagram (WK)

Fig. 3 shows the WK diagram for the 200hPa zonal wind field. Figs. 3(a), (b) and (c) display the symmetric part of the spectrum, while Figs. 3(d), (e) and (f) present the antisymmetric part. The wavenumber-frequency spectrum has been computed from

the raw data set in panels (a) and (d), while the remaining panels show wavenumber-frequency spectra computed from the reconstruction of the zonal wind field by considering only the barotropic (panels b and e) and baroclinic (panels c and f) components. The barotropic component refers to the vertical modes $m = 1, 2, 3, 4$ and $5$, which exhibit a barotropic structure throughout the troposphere, while the baroclinic component refers to the remaining vertical modes ($m > 5$). Figs. 3(a) and (d) are similar to the wavenumber-frequency spectra of tropical disturbances that have been documented in the literature (e.g.,

Takayabu, 1994a; Wheeler and Kiladis, 1999; Kiladis et al., 2009), and are presented here for comparison purposes. From Fig. 3(a) one notices a significant spectral peak band following the linear dispersion relation of the Kelvin waves between wave-numbers 1 and 10. Another significant spectral peak observed in Fig. 3(a) refers to westward propagating synoptic-scale disturbances with wavenumbers 1-5, a spectral domain that corresponds to the barotropic Rossby-Haurwitz waves (e.g., Gehne and Kleeman, 2012). The strongest power signal of the symmetric part of the observed 200hPa zonal wind field spectrum occurs

at wave-number 1 with an intraseasonal time-scale, which might be associated with the Madden Julian Oscillation (MJO). It is also noticeable the significant spectral peak associated with large-scale ($k = 3 - 7$) westward inertio-gravity waves, along with some significant signals above the dispersion curves associated with Rossby waves. The antisymmetric part of the spectrum (Fig. 3d) shows a strong signal along the dispersion relation of the eastward propagating inertio-gravity[3] modes with zonal wavenumbers 3-10.

The most remarkable difference between the wavenumber-frequency spectrum presented in Figs. 3 (a) and (d) and those wavenumber-frequency spectra of outgoing long-wave radiation (OLR) presented in the literature refers to the peak at planetary-scale (wavenumber 1-2) having an intraseasonal time-scale associated with the MJO. While this spectral peak here is strongly

---

[3]The first antisymmetric eastward propagating inertio-gravity mode can also be thought of as the eastward branch of the continuum spectrum of the mixed-Rossby-gravity mode. See Kiladis et al. (2016) and Dias and Kiladis (2016) for a further discussion.





concentrated at planetary-scale, the corresponding intraseasonal peak associated with the OLR presented by Kiladis et al. (2009) is more elongated along the zonal wavenumber spectrum, showing a dispersion relation $d\omega(k)/dk \approx 0$. This difference might be attributed to the fact that the circulation associated with moist convection might contain less kinetic energy than the large-scale circulation, which is dominated by Rossby and Kelvin waves. In addition, as the OLR field is a proxy of convection activity, its spectrum should contain the multiple scales associated with the convection organization (see, for instance, Majda and Stechmann, 2009; Khouider and Majda, 2006; Khouider et al., 2012).

The wavenumber-frequency spectrum of the barotropic ($m = 1 - 5$) component of the 200hPa zonal wind field shows that the peak associated with westward propagating synoptic-scale disturbances with wavenumbers 1-5 and a period of $T \sim 5$ days observed in Fig. 3(a) is indeed associated with barotropic waves. On the other hand, the baroclinic component of the symmetric part of 200mb zonal wind spectrum (Fig. 3c) is largely dominated by a spectrum that follows the Kelvin waves, with a particularly strong spectral peak with $k = 1$ and a period of $T \sim 30$ days that may be attributed to the Madden Julian Oscillation. Regarding the antisymmetric component of the 200mb zonal wind WK spectrum, one notices that its baroclinc component (Fig. 3f) is very similar to the corresponding full spectrum (3d) and is dominated by a spectral peak following the dispersion relation of the eastward propagating inertio-gravity mode. The barotropic component, on the other hand, shows spectral peaks in different regions of the spectrum, including a narrow peak related to $k = 1$ westward propagating disturbances with intraseasonal timescale, possibly associated with the MJO. Spectral peaks with characteristic periods of $T = 5 - 10$ days and zonal wavenumbers $k = 1 - 5$ exhibiting westward propagation, as well as a spectral peak with a period of $T = 3 - 5$ days and wavenumber $k = 6 - 9$ exhibiting eastward propagation, are also observed.

In order to help us to interpret the results presented in Fig. 3, we present in Figure 4 the energy spectrum as a function of the zonal wavenumber $k$ and the vertical index $m$. From Fig. 4 one observes that the spectrum is dominated by disturbances with large spatial scales ($k = 1 - 5$) and a barotropic structure in the troposphere $m = 1 - 5$, which agrees with the spatial structure displayed in Figure 2 that exhibits most part of total energy concentrated in the subtropical jets.

## 4.2 Rossby modes

Figure 5 shows the wavenumber-frequency spectrum (WK diagram) computed from the normal mode decomposition of the 200mb zonal wind field that retains only the rotational modes, with all the vertical modes (Figs. 5a and d) and with only the barotropic (Figs. 5b and e) and baroclinic (Figs. 5c and f) components. Figs. 5(a), (b) and (c) display the symmetric part of the spectrum, while Figs. 5(d), (e) and (f) present the antisymmetric part. For the symmetric part of the full vertical mode spectrum, one notices in Fig. 5a an energy concentration on the wave-number 1 with eastward propagation that is associated with the global-scale circulation pattern related to the Madden Julian Oscillation (see, for instance, the MJO skeleton theory composed of global-scale Rossby and Kelvin waves presented in Majda and Stechmann (2009)). A significant spectral peak is also found on synoptic time-scales (3-6 day period) with westward propagation, which was proposed by Gehne and Kleeman (2012) to be due to barotropic Rossby-Harwitz wave activity. This spectral peak on synoptic-scale having westward propagation is more pronounced and more closely related to the barotropic Rossby wave dispersion relation in the symmetric WK spectrum evaluated from the barotropic component of the zonal wind field presented in Fig. 5b. From the symmetric



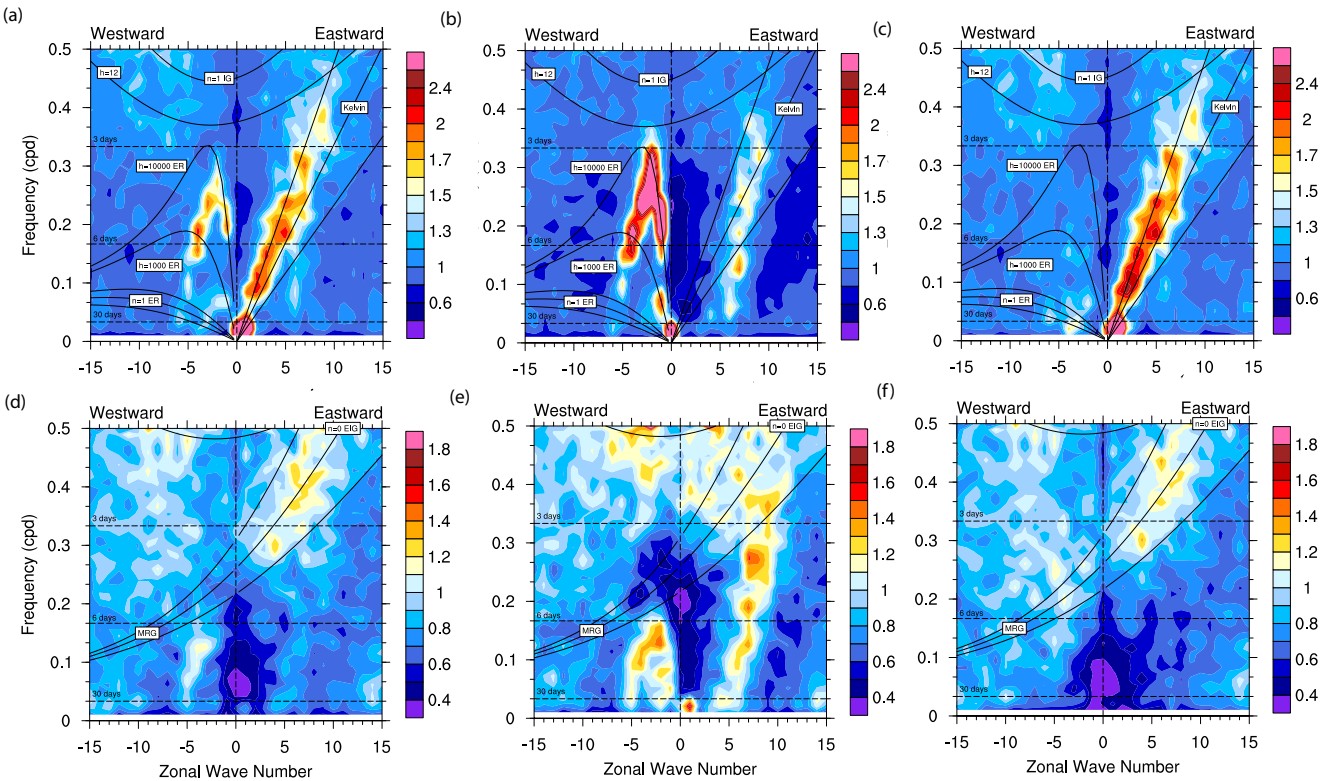

**Figure 3.** Frequency-wavenumber spectrum (Wheeler-Kiladis diagram) of the zonal wind field at 200hPa, for the symmetric (panels a, b and c) and antisymmetric (panels d, e and f) parts of the spectrum; panels (a) and (d) refer to the full spectrum of vertical modes, whereas panels (b) and (e) show the barotropic component and panels (c) and (f) the baroclinic component of the corresponding spectra.

baroclinic component displayed in Fig. 5c, one observes that the synoptic-scale spectral peak is significantly reduced by removing the barotropic waves, thus confirming the findings of Gehne and Kleeman (2012). On the other hand, the intense spectral peak at $k = 1$ associated with the MJO remains by removing the vertical modes with a barotropic structure in the

troposphere. This is in agreement with the fact that the global-scale circulation associated with the MJO is characterized by a tropospheric baroclinic structure (Madden and Julian, 1972). The antisymmetric part of the spectrum (Fig. 5d) shows a signal roughly following the dispersion relation of the MRG mode, which is also evident in the baroclinic part of this spectrum (Fig. 5f). The wavenumber-vertical mode spectrum of the rotational modes is presented in Figure 6, which is very similar in structure with the spectrum without any modal filtering presented in Figure 4, with the energy being concentrated in low zonal

wavenumbers and low vertical indices. It is also possible to note a secondary peak with a baroclinic structure in the troposphere associated with the vertical modes $m = 7 - 10$.



**Figure 4.** Energy spectrum of the atmospheric oscillations as a function of the zonal wavenumber and the vertical mode index.



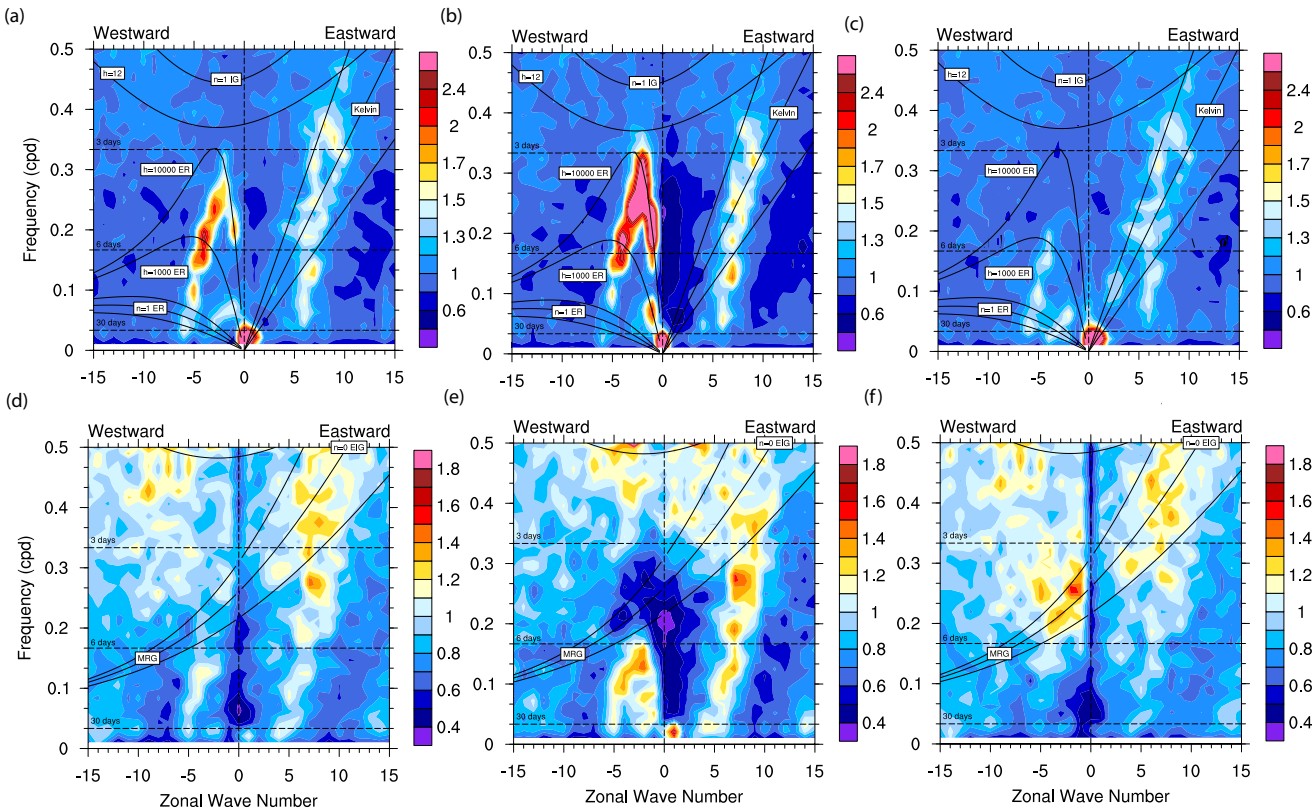

**Figure 5.** Frequency-wavenumber spectrum (Wheeler-Kiladis diagram) of the 200hPa zonal wind field associated with the rotational modes. Panels (a), (b) and (c) refer to the symmetric part, while panels (d), (e) and (f) refer to the antisymmetric part. The full spectrum of vertical modes is presented in panels (a) and (d); the barotropic component of the corresponding parts is displayed in panels (b) and (e), whilst the baroclinic component is illustrated in panels (c) and (f).

### 4.3 Westward inertio-gravity (WIG) modes

Now we present the Wheeler-Kiladis diagram computed from the contribution of unbalanced modes for the corresponding zonal wind field. Fig. 7 presents the wavenumber-frequency spectrum (WK diagram) computed from the normal mode decomposition of the 200mb zonal wind field that retains only the westward inertio-gravity (WIG) modes, including all the vertical modes (Figs. 7a and d) and only the barotropic (Figs. 7b and e) and baroclinic (Figs. 7c and f) components. Figs. 7(a), (b) and (c) display the symmetric part of the spectrum, while Figs. 7(d), (e) and (f) present the antisymmetric part. The WIG modes are particularly important since they have been evoked to explain the large-scale envelope of the MJO as a result of the interaction between WIG and eastward inertia-gravity (EIG) waves (see, for instance, Yang and Ingersoll, 2013). The WK diagram obtained from the WIG mode contribution to the zonal wind field displayed in Fig. 7 reveals some surprising aspects of the WIG wave propagation that strongly departs from the free linear theory of WIG wave propagation. Indeed, from Fig.

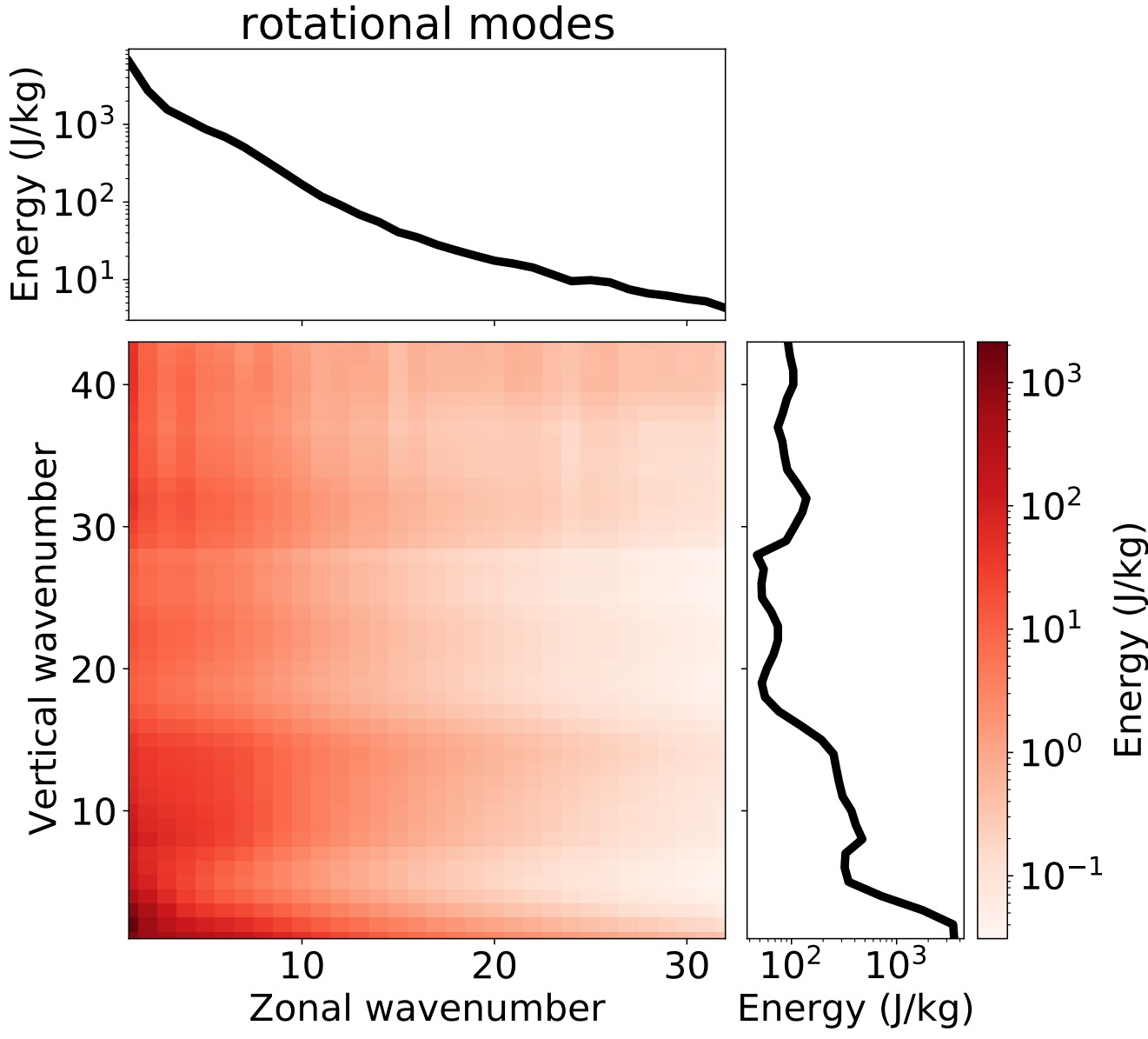

**Figure 6.** Energy spectrum of the rotational mode component of the atmospheric oscillations as a function of the zonal wavenumber and vertical mode.





7a it is noted the symmetric part of the spectrum exhibiting a strong spectral peak in the vicinity of the WIG wave dispersion relation for wave-numbers $k = 1 - 7$, in agreement with the linear theory. However, strong spectral peaks are also found along the dispersion curve of the Kelvin waves with zonal wavenumbers 4-14. This result might be an indicative of coherent non-
linear interaction and wave synchronization involving Kelvin and WIG waves. This theoretical mechanism involving coherent nonlinear interaction through phase synchronization has been explored in the contexts of plasma physics (Chian et al., 2010) and solar physics (Raphaldini et al., 2020a). Other significant spectral peaks are found along the dispersion relation of Rossby waves for $k = 1 - 6$. Fig. 7 also reveals other spectral peaks centered in the intraseasonal timescale, which is probably associated with the Madden-Julian Oscillation. The baroclinic component of the symmetric part of the WIG wave spectrum presented
in Fig. 7c largely resembles the spectrum presented in Fig. 7a, suggesting that the baroclinic component of the WIG waves is the dominant one, possibly due to the strong coupling of these waves with moist convection (Yang et al., 2003). In addition, the significant spectral peak of WIG waves with eastward propagation and a typical period within the intraseasonal timescale ($T \sim 30$ days) is suggestive of the role of the WIG waves in the Madden-Julian Oscillation. The role of gravity waves in the MJO is highlighted in a number of theories, such as the gravity wave theory of the MJO (Yang et al., 2007; Yang and Ingersoll,
2013), and the multi-cloud theory of convection parameterization documented in Khouider and Majda (2006) and Khouider and Majda (2007).

The power spectrum of the antisymmetric part, on the other hand, is much weaker, as can be observed in Fig. 7d for the full vertical mode spectrum as well as for its decomposition into barotropic (Fig. 7e) and baroclinic (Fig. 7f) components.

The wavenumber-vertical mode spectrum of the WIG waves (Figure 8) differs from that of the rotational modes, since the
primary peak in the vertical mode index is characterized by a baroclinic structure in the troposphere ($m = 6 - 9$). This result may be expected, since the divergent modes are associated with moist convection.

### 4.4 Eastward inertio-gravity (EIG) modes

The wavenumber-frequency spectrum computed from the contribution of eastward inertio-gravity (EIG) waves for the 200hPa zonal wind field is presented in Fig. 9. In this normal mode reconstruction of the zonal wind field, the Kelvin waves are not
considered as its contribution will be analysed separately in the next item of this section. As in Figs. 4, 5 and 7, panels a, b and c (d, e and f) display the symmetric (antisymmetric) part of the spectrum. In addition, panels (a) and (d) refer to the calculations with the full set of vertical modes, while panels (b) and (e) (c and f) display the results of the calculations with only the vertical modes $1 \leq m \geq 5$ ($m \geq 6$). The symmetric part of the spectrum (Figure 9a) shows some isolated spectral peaks near the linear dispersion curve of planetary-scale EIG modes. The symmetric part of the spectrum for the barotropic EIG modes (Figure
9b) presents a strong and broad peak on the Kelvin wave dispersion curve within the zonal wavenumber range of $k = 4 - 8$; the same signal extends toward the EIG dispersion curve for $k = 8$. There is also a spectral peak at the WIG wave dispersion curve at $k = 7 - 9$. The spectrum of the baroclinic EIG waves (Figure 9c) is quite similar to the full EIG spectrum (Figure 9a). Its analysis shows some moderate peaks close to the dispersion relation of the Kelvin waves. This suggests that some type of nonlinear interaction between EIG and Kelvin modes may occur, for instance with Kelvin waves exciting EIG waves as it
propagates. Narrower regions of significant power are also noticeable in the spectrum along the linear dispersion relation of



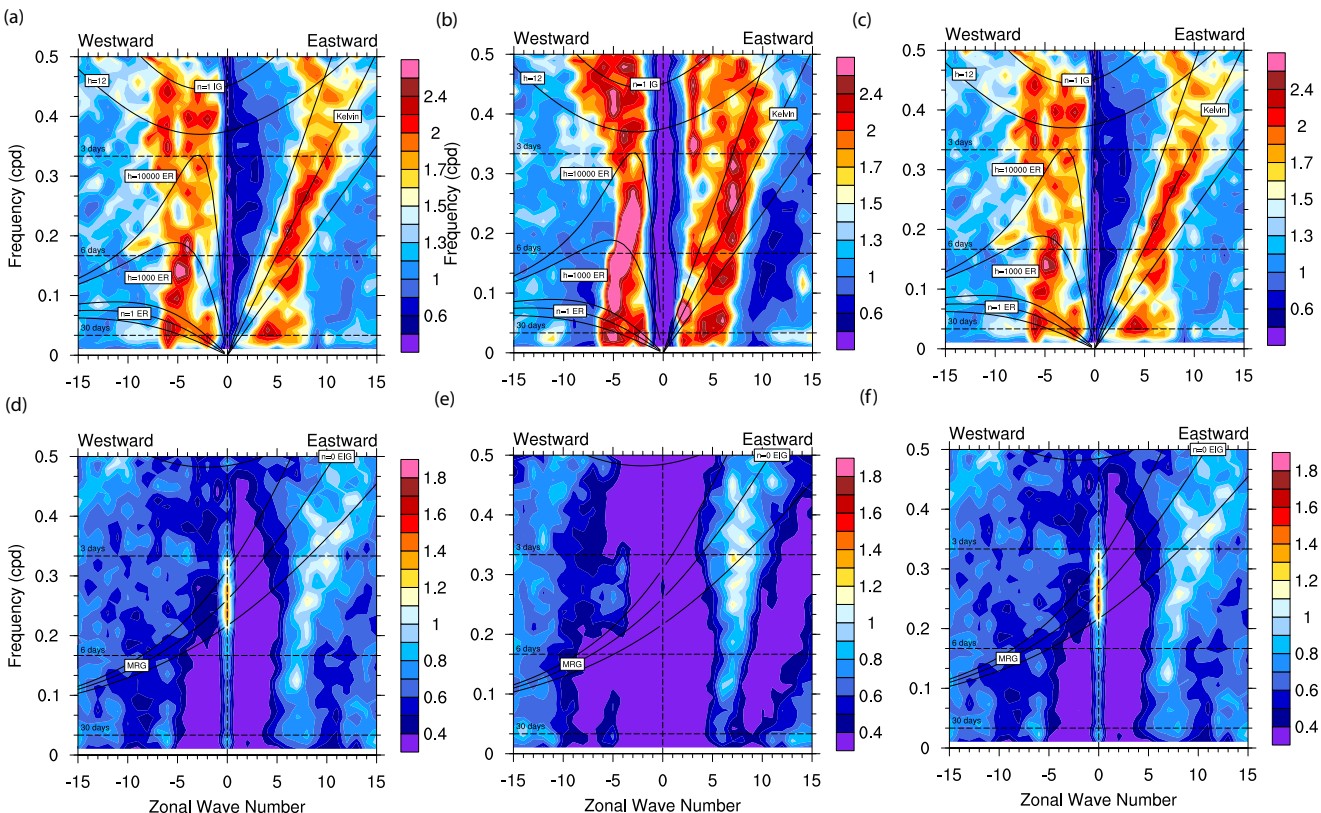

**Figure 7.** Frequency-wavenumber spectrum (Wheeler-Kiladis diagram) of the 200hPa zonal wind field associated with the westward inertio-gravity modes. Panels (a), (b) and (c) refer to the symmetric part, while panels (d), (e) and (f) refer to the antisymmetric part. The full spectrum of vertical modes is presented in panels (a) and (d); the barotropic component of the corresponding parts is displayed in panels (b) and (e), whilst the baroclinic component is illustrated in panels (c) and (f).

the EIG modes for low wavenumbers, $k = 1 - 4$, along with a moderate peak around the 30-day period close to the dispersion relation of the equatorial Rossby waves at high wavenumbers. For the antisymmetric part of the spectrum of the EIG modes, one observes a region of high power peak following the dispersion relation of the first antisymmetric EIG mode with low wavenumbers, $k = 1 - 5$.

Therefore, the results displayed in Fig. 9, in general, suggest that the EIG modes seem to be slave modes whose propagation is determined by other waves, such as the Kelvin and Rossby modes. The wavenumber-vertical mode spectrum of the EIG modes presented in Figure 10 is similar to that of the WIG modes, with a primary peak in the vertical mode index around $m = 6 - 10$ and the energy decaying with the zonal wavenumber.

There is a long standing problem in tropical dynamics to understand the lack of strong and broad spectral power along the dispersion relation of the EIG modes other than the first antisymmetric one, for instance when observed from the OLR field (Wheeler et al., 2000). The results presented in this section show that the spectral peaks on the symmetric part are weak (or,

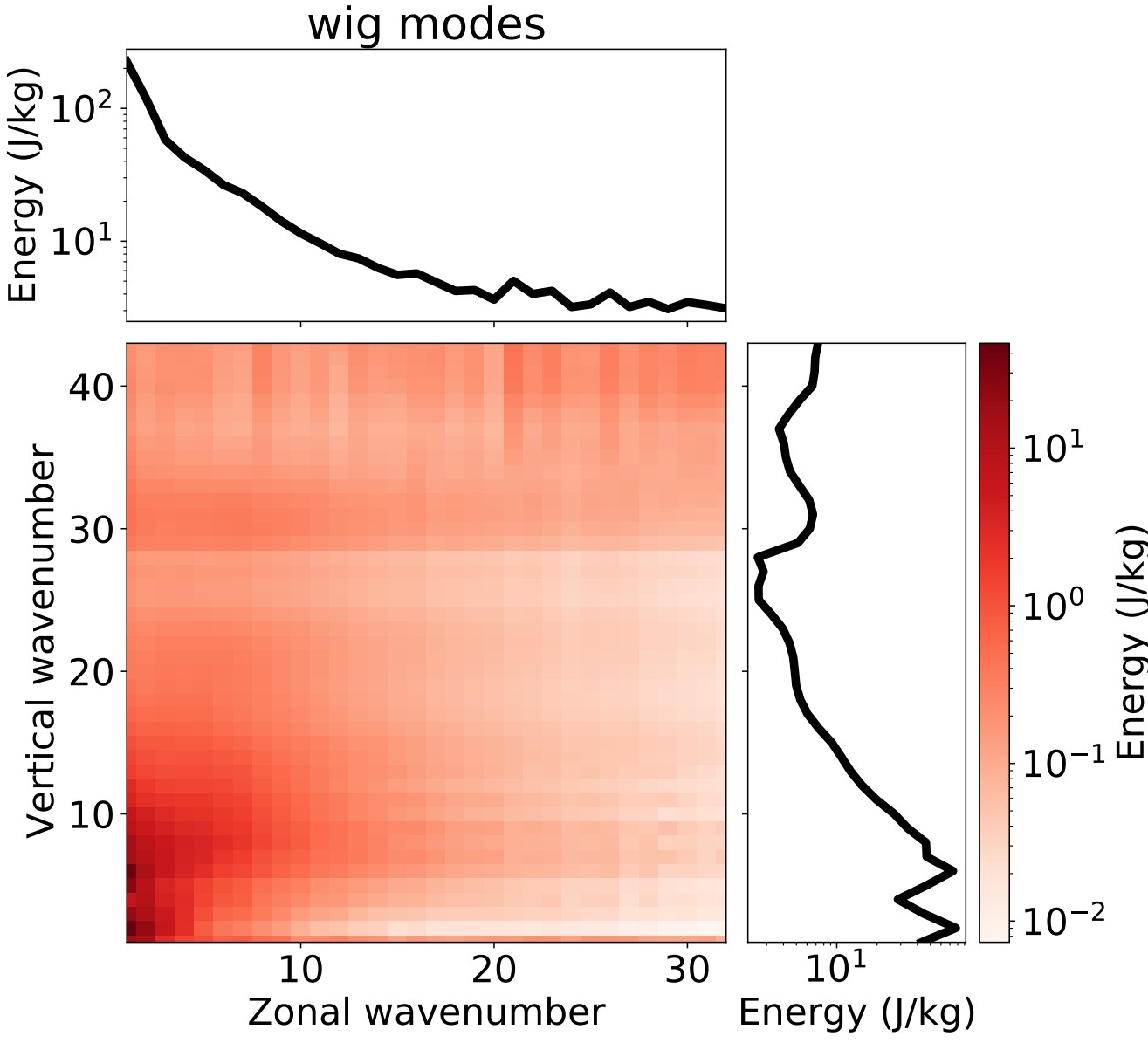

**Figure 8.** Energy spectrum of the westward inertio-gravity mode component of the atmospheric oscillations as a function of the zonal wavenumber and vertical mode.





### 4.5 Kelvin modes

The wavenumber-frequency spectrum computed from the contribution of the Kelvin waves for the 200hPa zonal wind field is presented in Fig. 11. Fig. 11a refers to the calculations with the full set of vertical modes, while Figs. 11b and c refer to the spectral reconstruction of the zonal wind field with only the barotropic ($1 \leq m \geq 5$) and baroclinic ($m \geq 6$) components, respectively.

Fig 11a shows a narrow peak having a high power around the zonal wavenumber $k = 1$ and the intraseasonal timescale 315 (30-70 days). This observed spectral peak is probably a manifestation of the role of the Kelvin mode in the morphology of the Madden-Julian Oscillation. Žagar et al. (2015), for instance, showed that the zonal wavenumber-1 Kelvin wave has an important contribution to the MJO. Recently, Raphaldini et al. (2020b) suggested that among the dominant planetary-scale waves of the MJO, the Kelvin mode is the first to be excited, transferring energy to the Rossby modes afterwards. In addition, one notices from Fig 11a a significant power in the intraseasonal range of frequencies for larger wavenumbers ($k = 9 - 15$), 320 suggesting that these Kelvin modes also significantly contribute to the MJO envelope. Another point to be noticed in Fig. 11 is that the Kelvin waves with intermediate wavenumbers ($k = 3 - 9$) do not seem to have a strong contribution to the MJO. It is also observed regions of strong power along the linear dispersion relation of the Kelvin waves, specially for wavenumbers 5-15. The most striking point on the results is regarding the spectral peaks on Kelvin waves with different frequencies, for instance, the ones oscillating in synoptic timescales (4-7 days) with wavenumbers $k = 12 - 15$ and eastward propagation (Fig. 325 11b). The most surprising result refers to the presence of strong power in the westward part of the spectrum, which appears to be predominantly due to the baroclinic part of the spectrum. We hypothesize here that it may be an indication of strong turbulent behavior associated with highly energetic convective events related to the Kelvin waves. Another important aspect of Kelvin waves that may be the origin of strong turbulence, and therefore a spectrum that largely departs from the linear theory, is that the Kelvin waves are non-dispersive and, consequently, their nonlinear dynamics may be described by a Burger equation, 330 which admits wave breaking and shock formation in a finite time (see, for instance, Boyd, 1980; Ripa, 1982; Boyd, 1998).

As in the corresponding spectra of the inertio-gravity waves, the wavenumber-vertical mode spectrum of the Kelvin waves (Fig. 12) shows a primary peak for vertical indices associated with a baroclinic structure in the troposphere $m = 6 - 10$, a signature of the important role of these waves in tropical convection (Fig. 11c).

### 4.6 Mixed Rossby-Gravity (MRG) waves

The wavenumber-frequency spectrum computed from the contribution of MRG waves to the 200hPa zonal wind field is illustrated in Fig. 13. It is shown that the spectrum is dominated by a narrow and elongated peak with wavenumbers $k = 1 - 2$ and characteristic timescale of $T\ 2-5$ days, with an westward propagation, and a secondary peak with eastward propagation around wavenumbers $k = 5 - 10$ (Fig. 13a). The spectra computed from the corresponding barotropic and baroclinic components are similar, however the barotropic component shows a more intense concentration of the power spectrum around these regions



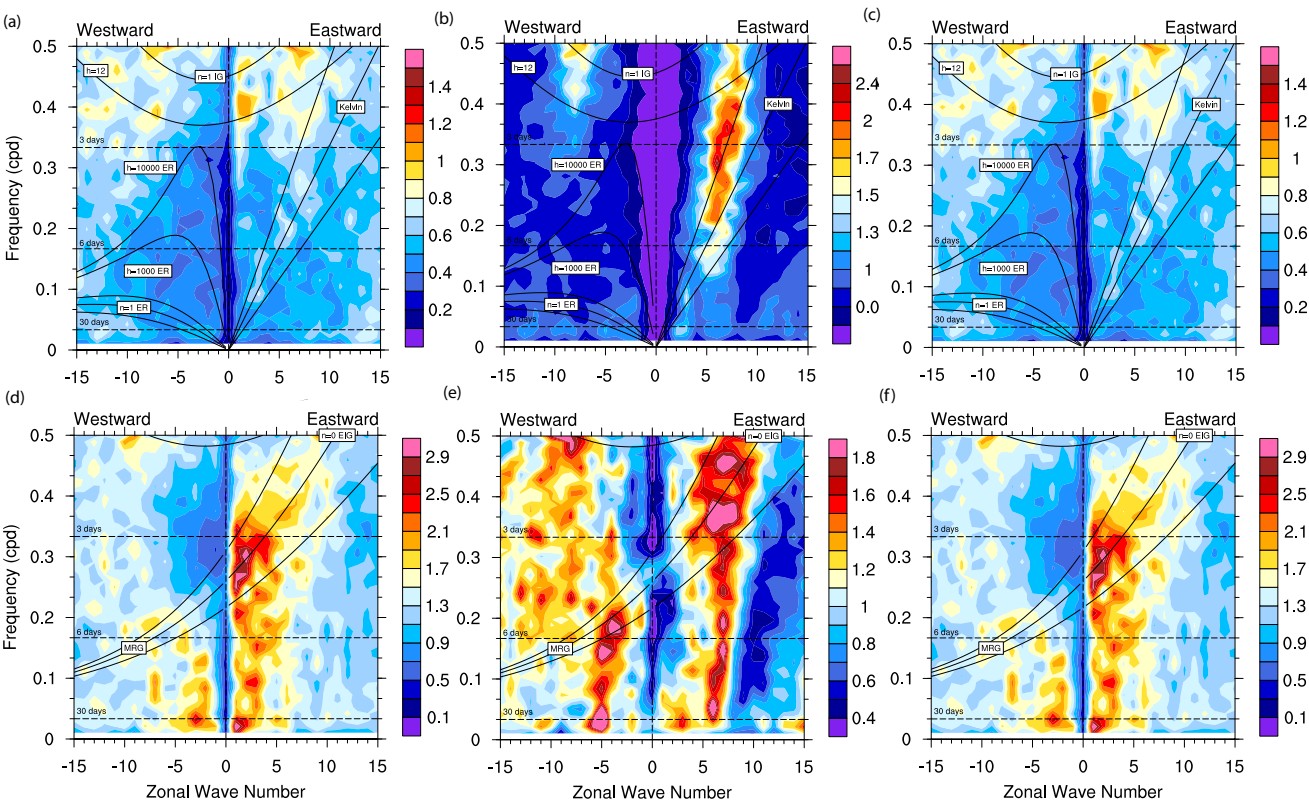

**Figure 9.** Frequency-wavenumber spectrum (Wheeler-Kiladis diagram) of the 200hPa zonal wind field associated with the eastward inertio-gravity modes. Panels (a), (b) and (c) refer to the symmetric part, while panels (d), (e) and (f) refer to the antisymmetric part. The full spectrum of vertical modes is presented in panels (a) and (d); the barotropic component of the corresponding parts is displayed in panels (b) and (e), whilst the baroclinic component is illustrated in panels (c) and (f).

and also an elongation of these structures toward lower frequencies (Fig. 13b). The wavenumber-vertical mode spectrum (Fig. 14) of the MRG modes shows a strong dominance of the barotropic structure in the troposphere $m = 1 - 4$, with a slowly decaying spectrum up to $k$ 8 in the zonal direction, along with a stepper slope for larger wavenumbers.

## 5   Discussion

In the present study we have analysed the space-time spectrum of equatorial disturbances by computing the wavenumber-
frequency spectrum of normal mode decomposed dynamical fields obtained from the Era-Interim (ERAI) reanalysis data. In this approach, the large-scale atmospheric dynamical fields are projected onto the normal mode functions defined as the eigensolutions of the compressible primitive equations in spherical coordinates, linearized around a resting background state. From filtered versions of the spectral reconstruction of these dynamical field variables considering only a single mode type

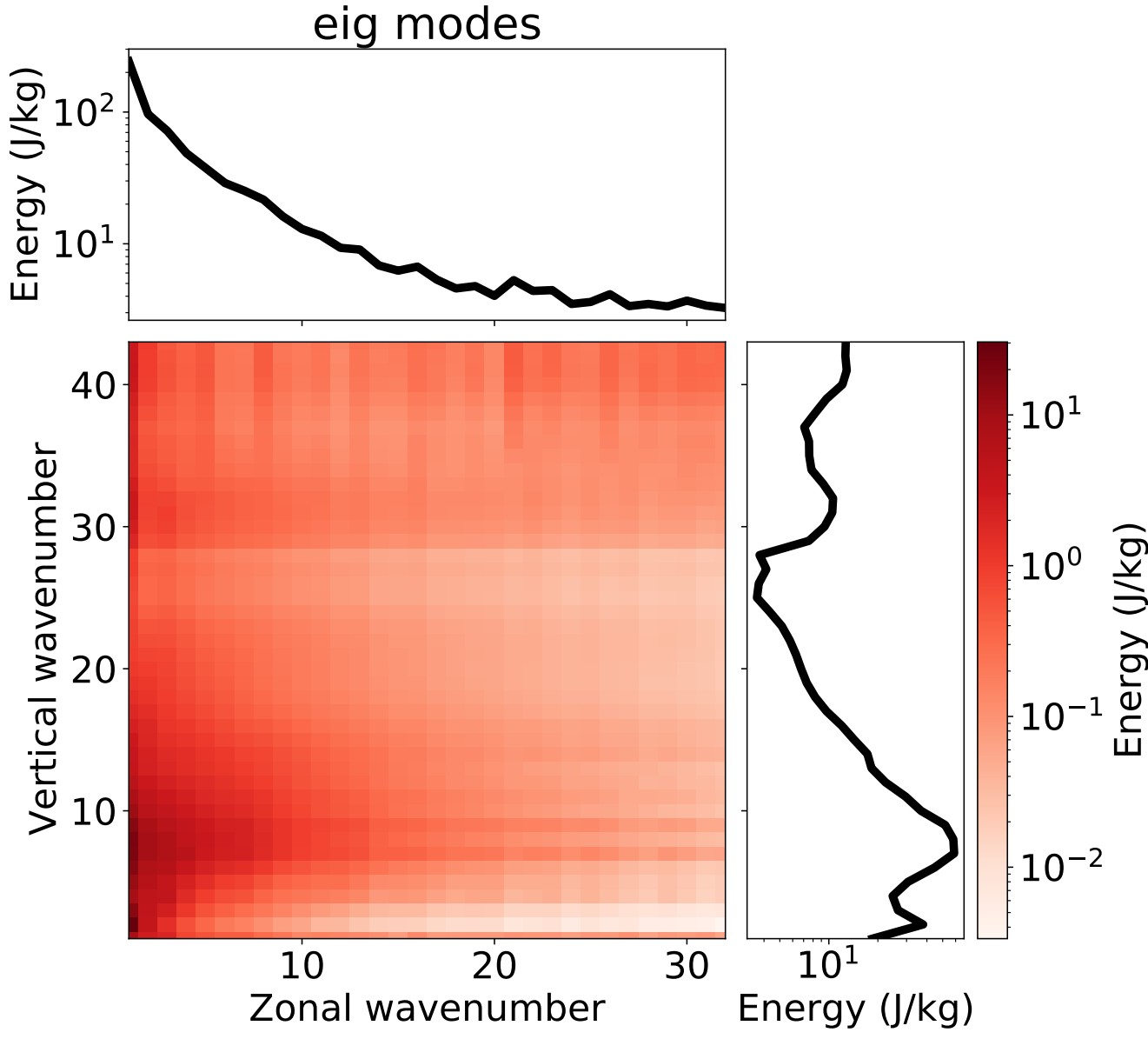

**Figure 10.** Energy spectrum of the eastward inertio-gravity wave component of the atmospheric oscillations as a function of the zonal wavenumber and vertical mode.



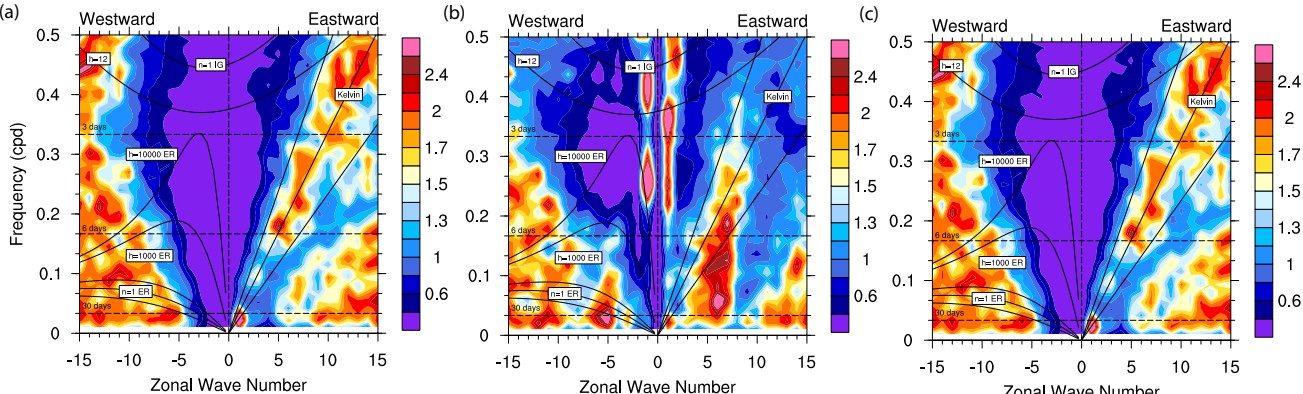

**Figure 11.** Frequency-wavenumber spectrum (Wheeler-Kiladis diagram) of the 200hPa zonal wind field associated with the Kelvin modes. The full spectrum of vertical modes is presented in panel (a), while panels (b) and (c) refer to the barotropic and baroclinic components, respectively.

(Rossby, Kelvin, WIG, EIG, and MRG modes), as well as their tropospheric barotropic and baroclinic components, we have
computed the wavenumber-frequency spectrum using the methodology proposed by Wheeler and Kiladis (1999) for the 200hPa zonal wind field. Unlike other studies in the literature (e.g., Wheeler and Kiladis, 1999; Castanheira and Marques, 2015; Takayabu, 1994a, b) that analysed the OLR spectrum to focus on the coupling of the waves with moist convection, here we analysed a dynamical field variable to primarily investigate the departures of the wave spectra from that predicted by the linear theory.

Our results show some aspects that agree with the linear theory, for instance, the barotropic Rossby modes showing a spectral peak that clearly follows the dispersion relation according to the linear theory. In addition, part of the Kelvin mode power spectrum follows what is predicted by the linear theory, with an elongated spectral peak, seemingly a non-dispersive character, having an eastward propagation. Part of the power spectrum obtained for the WIG modes, corresponding to wavenumbers $k = 1 - 7$, is shown to follow the linear dispersion relation as well.

On the other hand, we also found some spectral peaks that largely departure from what is expected from the linear theory. First, for Kelvin waves, we verified a rather complex distribution of the energy throughout the wavenumber-frequency spectrum, especially for large wavenumbers. A possible explanation for this result stems from the non-dispersive nature of the Kelvin wave, which imply a strong nonlinear coupling among all of its harmonics, which may lead to wave breaking and shock formation in a finite time (see, for instance, Boyd, 2018, and references therin), and therefore a strong turbulence. Similarly,
the obtained wavenumber-frequency spectrum for the inertio-gravity waves show a very peculiar phenomenon: their propagation seem to be "slaved" by other modes. For instance, the power spectrum obtained for the EIG modes, specially for their barotropic component, clearly follows the dispersion relation of the Kelvin waves, while the power spectrum obtained for the WIG modes suggests a propagation along the dispersion relation of the equatorial Rossby waves, barotropic Rossby waves,

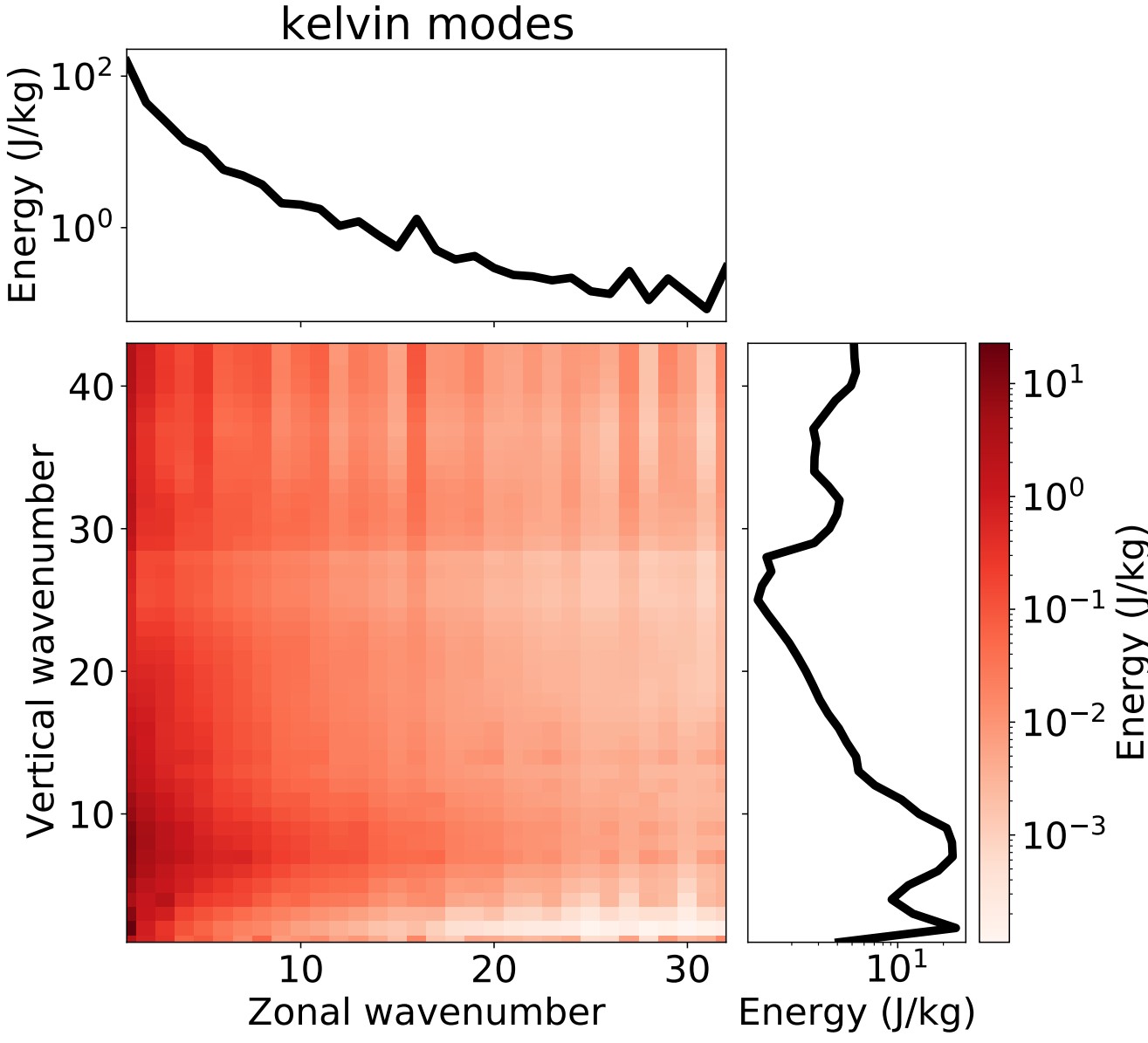

**Figure 12.** Energy spectrum of the Kelvin mode component of the atmospheric oscillations as a function of the zonal wavenumber and vertical mode.



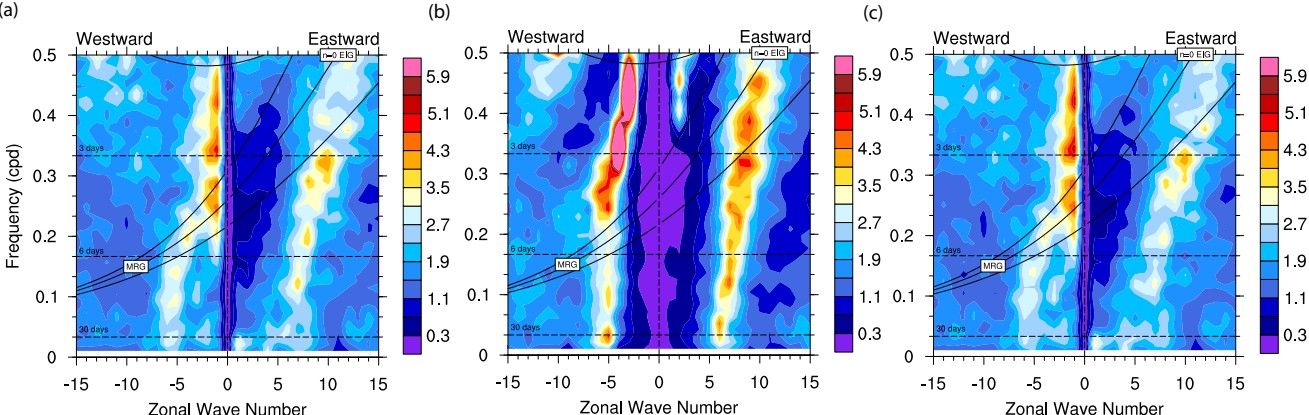

**Figure 13.** Frequency-wavenumber spectrum (Wheeler-Kiladis diagram) of the 200hPa zonal wind field associated with the mixed Rossby-gravity modes. The full spectrum of vertical modes is presented in panel (a), while panels (b) and (c) refer to the barotropic and baroclinic components, respectively.

MJO and Kelvin waves. These results suggest a synchronization effect of the inertio-gravity modes with other, possibly more
energetic, modes as documented in some of the MJO theories (see, for instance, Yang and Ingersoll, 2013, 2014).

Another important result refers to the analysis of EIG mode power spectra that sheds light on the question of the absence of
strong and broad spectral power peaks in the vicinity of the dispersion relation of EIG modes (except the first antisymmetric
one that is a continuum eastward extension of the MRG modes) on the Wheeler-Kiladis diagram. This fact was first highlighted
by Takayabu (1994b) and then by the subsequent work of Wheeler et al. (2000) on the OLR field. Here, we showed that there
are weak (or, at most, moderate) and isolate power peaks along the dispersion relation of these EIG modes. In contrast, the
barotropic component of these EIG modes depicted significant energy following the dispersion relation of the Kelvin waves,
whereas weak spectral peaks are observed on the baroclinic component, which is the one that mostly contribute to convection
and therefore to the OLR field. Further theoretical investigations need to be done in the direction of understanding the lack of
strong spectral peaks on the wavenumber-frequency spectrum associated with the baroclinic EIG modes and on the departure
of the barotropic EIG W-K diagram from the linear dispersion relation.

Recently, some studies (Rostami and Zeitlin, 2019, 2020) revisited the problem of the equatorial wave geostrophic ad-
justment in the presence of moist-convective processes. In particular, Rostami and Zeitlin (2020), with a two-layer moist-
convective nonlinear shallow water model, showed that Kelvin and Rossby modes might fuse into a hybrid structure, linked
through a region of enhanced condensation, and this region is related to westward inertio-gravity waves. These results suggest
that a combination of moist-convective and nonlinear processes might explain the spectral peaks associated with equatorial
Rossby waves, barotropic Rossbys waves, MJO, and Kelvin waves found in the observed wavenumber-frequency spectrum of
the westward inertio-gravity wave field.

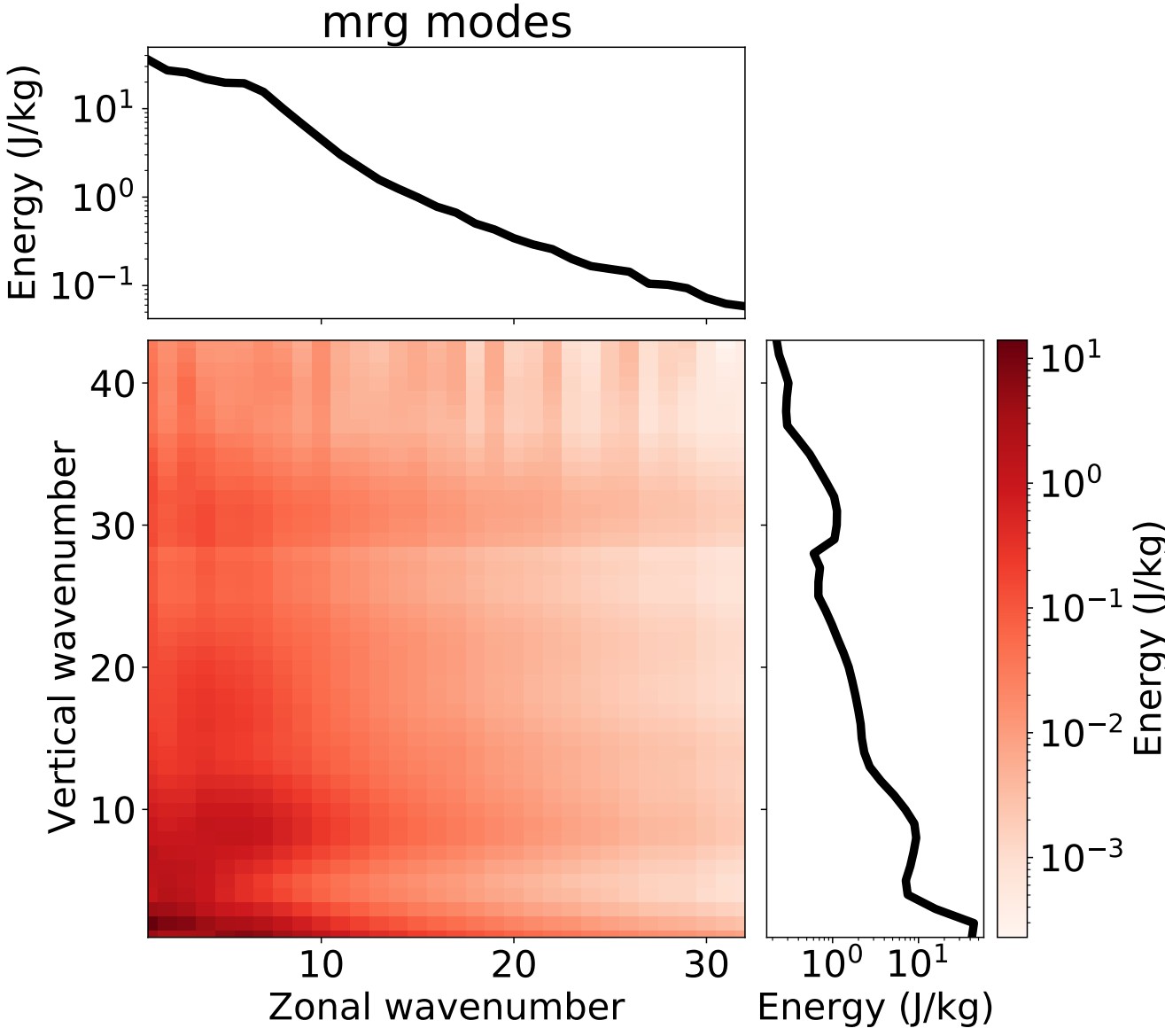

**Figure 14.** Energy spectrum of the mixed Rossby-gravity mode component of the atmospheric oscillations as a function of the zonal wavenumber and vertical mode.



It is a common practice in the studies of tropical dynamics to filter the field variables within the spectral range associated
with the observed peaks along the dispersion relation of a certain linear eigenmode and to attribute the resulting filtered field
as being due to the activity of that corresponding eigenmode (see, for instance, Wheeler et al., 2000; Kiladis et al., 2009).
Our results show that this practice must be used with caution, since we have shown some observed spectra computed from
the contribution of a certain mode type to the 200hPa zonal wind field exhibiting peaks close to the dispersion relation of
another eigenmode. One example refers to the wavenumber-frequency spectrum computed from the contribution of westward
and eastward inertio-gravity waves displaying spectral peaks along the dispersion relation of the Kelvin waves. Furthermore,
a significant portion of the computed WIG mode spectrum showed peaks along the dispersion relation of equatorial Rossby
waves. These results suggested that gravity waves often have their propagation "slaved" to other modes. We argue a possible
physical explanation for this phenomenon including a nonlinear wave synchronization and coherent wave interaction that have
been reported in the literature of astrophysical magnetohydrodynamical flows (Raphaldini et al., 2020a; Chian et al., 2010;
Miranda et al., 2015).

*Author contributions.* "AT and BR designed the data analysis and AT carried them out. AT and all co-authors analyzed and discussed the
results. AT prepared the manuscript with contributions from all co-authors."

*Competing interests.* The authors declare that they have no conflict of interest.

*Acknowledgements.* The work reported here has been supported by Fundação de Amparo à Pesquisa do Estado de São Paulo (FAPESP)
through grant 15/50122-0.



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
