# Peer review of "Observed wavenumber-frequency spectrum of global, normal mode function decomposed, fields: a possible evidence for nonlinear effects on the wave dynamics"

_Weather and Climate Dynamics, 2021_

## Author Comment (AC1)

Response to Reviewer 1's report regarding the paper WCD MS No. wcd-2021-21: "Observed wavenumber-frequency spectrum of global, normal mode function decomposed, fields: a possible evidence for nonlinear effects on the wave dynamics", by André Seiji Wakate Teruya, Breno Raphaldini, Victor C. Mayta, Carlos F. M. Raupp, and Pedro L. da Silva Dias.

**June 2021**

In this reply we provide some preliminary responses to the main issues raised by Reviewer 1. Reviewer 1 raised some important questions, and we are confident to be able to address all of them. Also, we recognize that the incorporation of the revisions discussed here on the manuscript will significantly improve its quality. We provide here some elements on the theory regarding the nonlinear correction of the linear eigenfrequencies, the so-called Stokes correction. We were planning to evoke this theory to explain the observational results presented in the present paper in a follow up article. However, in face of the comments of Reviewer 1, we feel appropriate to introduce the Stokes correction theory here. We think that by further discussing a specific nonlinear mechanism to explain the departures of the observed time frequencies from their expected values predicted by the linear theory might make more clear the original contribution of the present paper.

The revisions discussed here are not to be considered as the final reply to all comments; we still plan to elaborate more on them, particularly on the statistical tests on the coherence analysis. The intention here is to show that we are able to provide a satisfactory answer to the main issues, while other minor issues will be addressed in the final answer, should we be given the opportunity to do so.

1. "My main comment on the results is about the lack of discussion of the eastward-propagating signals in WK diagrams using 200 hPa wind from the westward-propagating linear modes, and vice versa."

**Answer:** The reviewer pointed out the lack of discussion about eastward-propagating signals in WK diagrams from westward-propagating linear modes and vice-versa. So, to improve this discussion, we will include a case study of eastward-propagating signals associated with westward inertia-gravity waves and westward-propagating signals associated with Kelvin waves (an eastward-propagating linear eigenmode).

For both cases, we show the Hovmöller diagram associated with the 200hPa zonal wind field, averaged from 10S to 10N, due to Kelvin and westward inertia-gravity (WIG) waves (Figure 4). The signal of each wave was filtered in space and time by removing signals with periods bigger than 10 days keeping only signals with zonal wavenumbers between 4 and 6. Figure 4 shows eastward and westward propagating signals for both Kelvin and WIG waves.

2. "For example, how can one understand results of section 4.6 and figure 13 on mixed Rossby-gravity waves? The authors say that they analyzed the zonal wind at 200 hPa, which is expected to be rather small for the mixed Rossby-gravity mode, but the amplitude of its variance in figure 13 exceeds the variance in any other plot, including the total zonal wind. I assume that the same scaling is applied in all WK diagrams, as the amplitudes are not discussed and colour bars not explained, but I may be wrong. These things should be paid attention to."

Answer: The reviewer pointed out that the zonal wind field associated with mixed Rossby-gravity waves is rather small and should not be used to characterize them. This is completely accurate and we plane to address this issue by calculating the symmetric component of the WK diagram with a 96-day window, and with 65 overlapping days, for the 200hPa meridional wind field (Figures 1, 2 and 3). We also intend to clarify that each WK diagram is obtained by removing the background spectrum from the raw spectrum of each case [1]. This procedure is used to allow the identification of equatorial wave disturbances present in the raw spectrum of each normal mode of the analyzed fields, but the amplitude of these normalized spectra are not comparable to each other.

3. "For a similar result with westward gravity modes, the authors write in the final discussion section that 'a combination of moist-convective and nonlinear processes might explain the spectral peaks associated with equatorial Rossby waves, barotropic Rossby waves, MJO, and Kelvin waves found in the observed wavenumber-frequency spectrum of the westward inertio-gravity wave field." **Answer:** In order to further improve the discussion regarding how a combination of moist-convective and nonlinear processes might explain some of the observed spectral peaks, we will include a section in the manuscript about this result. A draft of this section is shown in Section 1 of the present reply.

4. "Similarly, 'gravity waves often have their propagation "slaved" to other modes'. This is appropriately a final discussion section, as there are no clear new results or explanation why WK diagrams look the way they do in this case"

**Answer:** The reviewer also pointed out a lack of discussion regarding the strong spectral peak on WIG waves' WK diagrams. To complement this discussion, we plotted the Hovmöller diagrams on Figure 4 showing WIG waves propagating as Kelvin waves in the second half of January 2014. We also calculated the spatial coherence between WIG and Kelvin waves' constributions to the zonal wind field (Figure 5). This spatial coherence is defined as,

$$\mathscr{C}(\sigma) = \frac{|\mathscr{F}[R_{u_1, u_2}(\tau)]|^2}{\mathscr{F}[R_{u_2, u_2}(\tau)]\mathscr{F}[R_{u_1, u_1}(\tau)]},\tag{1}$$

where  $\tau$  is a time lag and

$$R_{u_1,u_2}(\tau) = \frac{1}{2\pi T} \int_0^T \int_0^{2\pi} \overline{u_1(\theta,t)} u_2(\theta,t-\tau) d\theta dt,$$
(2)

and

$$\mathscr{F}[f(\tau)](\sigma) = \int_{-\infty}^{\infty} f(\tau) e^{-2i\pi\sigma\tau} d\tau.$$
(3)

Figure 5 displays a strong spatial coherence between Kelvin and WIG waves at the synoptic timescale.

**1 Nonlinear modification of normal mode frequency**

Current theories of convectively coupled equatorial waves attempt to explain the modification of the propagation speed of a particular eigenmode due to its interaction with moist convection. The basic idea is that the coupling between the largescale wave with moist convection reduces the effective static stability parameter of the atmosphere (in comparison with the corresponding value of a dry atmosphere), resulting in a slower propagation compared to what is expected from the normal mode theory of a dry atmosphere. However, apart from the coupling with moist convection, the nonlinearity can also reduce the wave propagation and, consequently, yield an observed wave spectrum that largely departs from what is expected from the linear theory of a dry atmosphere.

Here we introduce a possible nonlinear theoretical mechanism that can potentially modify the linear frequency of the waves, resulting in different propagation speeds. This mechanism refers to the so-called *Stokes correction* of the linear eigenfrequency [2, 3] and can be easily explained in the context of the nonlinear Schrodinger equation, an universal model for the propagation of dispersive wave packages in the presence of nonlinearity that was first introduced by [4, 5] in the context of atmospheric and oceanic waves. Similar models were proposed by [6, 7], for the generation of equatorial waves, and [8], for the Madden Julian Oscillation.

To introduce the mechanism in the simplest possible setting, let us consider the evolution equation for a monochromatic wave undergoing a nonlinear self-interaction. For a wave with amplitude A and eigenfrequency  $\omega(k)$ , the evolution equation for the spectral amplitude can be written as

$$\frac{dA}{dt} = i\omega(k)A + \gamma A|A|^2 \tag{4}$$

From the energy conservation (i.e., the Manley-Rowe relations), it follows that  $E = |A|^2$  is a constant, therefore one may write eq. (4) as:

$$\frac{dA}{dt} = i\tilde{\omega}(k)A\tag{5}$$

where  $\tilde{\omega} = \omega + \gamma |A|^2$ . From the equation above, one can see that the term  $\gamma |A|^2 = \gamma E$  consists of a nonlinear correction to the linear frequency  $\omega$ . More generally, for a system of nonlinearly interacting waves in which three-wave resonances are not possible (for instance, the intertio-gravity waves [9]), its evolution can be described by a set of ordinary differential equations with cubic nonlinearity. In this context, the dynamical equations for the evolution of a wave system may we written as

$$\frac{dA_k}{dt} = i\omega_k A_k + \sum_{k'} \sum_{k''} \sum_{k''} C_{k',k'',k'''}^k A_{k'} A_{k''} A_{k'''} + \sum_{k'} \gamma_{k'} A_k |A_{k'}|^2 \tag{6}$$

Again, from the conservation of the Manley-Rowe relations (see [10], sec. 4.5, and [11], chapter 7), equation (6) can be written as:

$$\frac{dA_k}{dt} = i\tilde{\omega}_k A_k + \sum_{k'} \sum_{k''} \sum_{k'''} C_{k',k'',k'''}^k A_{k'} A_{k''} A_{k''} A_{k'''}$$
(7)

where

$$\tilde{\omega}_k = \omega_k + \sum_{k'} \gamma_{k'} |A_{k'}|^2 \tag{8}$$

Now, unlike the monochromatic case, the frequency  $\omega$  of the wave is modified by its interaction with other waves as well. We note that although the equations governing the dynamics of geophysical fluids have quadratic nonlinearities, cubic terms in the wave amplitude evolution equations may become evident via near-idendity normal form transformations, in case three wave resonances are not possible [12, 13], or via asymptotic analysis [5, 14].

**References**

- Matthew Wheeler and George N Kiladis. Convectively coupled equatorial waves: Analysis of clouds and temperature in the wavenumber-frequency domain. *Journal of the Atmospheric Sciences*, 56(3):374–399, 1999.
- [2] Michael Stiassniea and Lev Shemerb. On the interaction of four water-waves. Wave motion, 41(4):307–328, 2005.
- [3] Raphael Stuhlmeier and Michael Stiassnie. Nonlinear dispersion for ocean surface waves. 2019.
- [4] John P Boyd. Equatorial solitary waves. part i: Rossby solitons. Journal of Physical Oceanography, 10(11):1699–1717, 1980.
- [5] John P Boyd. Equatorial solitary waves. part 2: Envelope solitons. Journal of Physical Oceanography, 13(3):428–449, 1983.
- [6] GM Reznik and V Zeitlin. Resonant excitation of rossby waves in the equatorial waveguide and their nonlinear evolution. *Physical review letters*, 96(3):034502, 2006.
- [7] G Reznik and V Zeitlin. Resonant excitation and nonlinear evolution of waves in the equatorial waveguide in the presence of the mean current. *Physical review letters*, 99(6):064501, 2007.
- [8] Jiansheng Zou and Han-Ru Cho. A nonlinear schrödinger equation model of the intraseasonal oscillation. Journal of the atmospheric sciences, 57(15):2435–2444, 2000.
- [9] Andrew Majda. Introduction to PDEs and Waves for the Atmosphere and Ocean, volume 9. American Mathematical Soc., 2003.
- [10] Elena Kartashova. Nonlinear resonance analysis: theory, computation, applications. Cambridge University Press, 2010.
- [11] Alex DD Craik. Wave interactions and fluid flows. Cambridge University Press, 1988.
- [12] Stephen Wiggins, Stephen Wiggins, and Martin Golubitsky. Introduction to applied nonlinear dynamical systems and chaos, volume 2. Springer, 1990.
- [13] Miguel Onorato, Lara Vozella, Davide Proment, and Yuri V Lvov. Route to thermalization in the  $\alpha$ -fermi-pasta-ulam system. Proceedings of the National Academy of Sciences, 112(14):4208-4213, 2015.
- [14] Mark J Ablowitz. Nonlinear dispersive waves: asymptotic analysis and solitons, volume 47. Cambridge University Press, 2011.